# Impact of Biogenic Structures of the Soil-Nesting Ants *Lasius niger* and *Lasius flavus* on the Soil Microarthropod Community in Urban Green Spaces

**DOI:** 10.3390/insects16101058

**Published:** 2025-10-17

**Authors:** Maria Sterzyńska, Dariusz J. Gwiazdowicz, Paweł Nicia, Paweł Zadrożny, Gema Trigos-Peral, Mohamed W. Negm

**Affiliations:** 1Museum and Institute of Zoology PAS, Twarda 51/55, 00-818 Warsaw, Poland; gtrigos@miiz.waw.pl; 2Department of Forest Entomology and Pathology, Poznań University of Life Sciences, Wojska Polskiego, 71c, 60-625 Poznań, Poland; dariusz.gwiazdowicz@up.poznan.pl; 3Department of Soil Science and Agrophysics, University of Agriculture in Kraków, Al. Mickiewicza 21, 31-120 Kraków, Poland; rrnicia@cyf-kr.edu.pl (P.N.); pawel.zadrozny@urk.edu.pl (P.Z.); 4Department of Plant Protection, Faculty of Agriculture, Assiut University, Assiut 71526, Egypt; waleednegm@yahoo.com

**Keywords:** soil disturbances, biodiversity, ant mounds, microarthropods, Collembola, Acari

## Abstract

**Simple Summary:**

Some animals, like ants, can change their surroundings in ways that affect other species. However, we still do not fully understand how these changes impact soil organisms. In this study, we explored how two common ant species, the black garden ant and the yellow meadow ant, influence the presence and diversity of soil microarthropods (both predators and detritus feeders) in urban soils. We compared soil from ant nests with nearby soil that had no ant activity. Our goal was to see if ant activity changes the number and types of these important soil organisms. The results showed that ant nests do affect both the abundance and the variety of these groups, with different responses depending on the role each group plays in the soil. Some groups remain unchanged, while others decreased in number. These findings show that ants act as ecosystem engineers by shaping who lives in the soil. Understanding these effects is important for protecting belowground biodiversity and can help guide the management of green areas in cities to support healthy and balanced ecosystems.

**Abstract:**

Organisms that physically modify their environment, known as ecosystem engineers, can influence resource availability, species interactions and the structure of soil communities. However, the specific effect of ecosystem engineers like ants on the abundance and diversity of non-engineering soil organisms remains understudied. To address this knowledge gap, we conducted a survey of a multi-taxon belowground community of soil microarthropods—Collembola, Mesostigmata, Oribatida and Actinedida—in urban areas, comparing nest mounds of the ant species *Lasius niger* and *Lasius flavus* with areas without ant-nesting activity (control). We hypothesised differences in abundance and distribution patterns of different soil microarthropod taxa between ant mounds and the control soil. We also hypothesised that ant-induced soil disturbance is species-specific, and may result in different patterns of diversity and composition of soil microarthropod assemblages within trophic levels, such as among detritivores (e.g., Collembola) and predators (e.g., Mesostigmata). Our results reveal how ecological filters shape different soil microarthropod groups’ responses to ant-driven changes in their environment. As we expected, soil disturbance caused by ant nest-building activity significantly influenced the abundance, distribution patterns and diversity of soil microarthropods, especially in the assembly of detritivorous—but not predatory—guilds of soil microarthropods.

## 1. Introduction

Organisms capable of physically engineering ecosystems have the capacity to directly or indirectly, and either positively or negatively, shape resource availability, species interactions, species diversity and abundance [1]. Soil engineers, as defined by [2], can directly or indirectly modify resource availability through bioturbation processes that actively alter the physical state of abiotic and biotic material [3], and create distinctly specific environmental conditions that differ from those of the surrounding soil [4,5]. Specifically, soil engineers perform continuous physical work on soils and sediments, displacing and transporting soil organic and mineral compounds, and creating functional biogenic structures such as organo-mineral aggregates (e.g., faeces, mounds, aggregates and gallery walls) and macropores (e.g., galleries and chambers). These structures have specific physical, chemical and biological properties [6], and are therefore considered to be important drivers of soil diversity and ecological filters (dispersal, abiotic and biotic factors which filter species into local communities), potentially influencing the occurrence and distribution of other soil species, including microarthropods.

Soil provides habitat for a wide range of organisms varying in size (from micro- to macrofauna), trophic status and ecological function [7,8,9]. These organisms contribute to the performance of essential soil-based ecosystem functions and services, such as litter fragmentation and decomposition, nutrient cycling, soil structuring and soil organic matter formation (e.g., [9,10,11]). Within the soil mesofauna, Collembola and Acari are the two most diverse and abundant groups of soil microarthropods inhabiting the upper soil horizons. As belowground consumers, they share a similar pool of resources and predators, and are members of soil micro-food webs defined by the detritivore and microbivore trophic guilds [12]. However, little is known about how soil engineers can influence the abundance and diversity of different trophic guilds among soil microarthropods.

Biogenic structures and/or microhabitats created by ecosystem engineers are ecologically important because they are considered to be resource and activity hotspots, patches or even islands in soil ecosystems (e.g., [4,13]). As a resource, biogenic structures not only modify the soil structure, but can also influence local community assembly processes that shape the identity and abundance of species in belowground communities [14]. It is widely recognised that the linkages between soil fauna and soil processes and ecosystem functioning depend on biodiversity, and are controlled by both inter- and intraspecific complex interactions, as well as bottom-up and top-down processes [15]. However, the links between the effects of soil engineers, the biodiversity of non-engineering soil-dwelling species, and ecological interactions in the soil have not been clearly and sufficiently identified.

Ants contribute significantly to soil biodiversity and belowground soil processes [16]. As they can physically engineer the soil ecosystem, they serve an important function as ‘facilitators’ or ‘regulators’ of biotic interactions in the soil [17]. They also support soil-based ecosystem services [18] and are one of the main sources of small-scale soil disturbance [19]. The soil-engineering activity of ants is based on digging to construct nest structures that include chambers in which to host the colony, and galleries and sheets that facilitate foraging activity both below and above the ground. It is generally accepted that the characteristics of the soil and the architecture of the nest structures allow ants to reduce environmental hazards (i.e., predation, temperature and moisture fluctuation) and optimise colony development [4,20,21]. On the other hand, these biogenic structures produced by ant engineering activity—usually referred to as the myrmecosphere [22]—alter soil physical and chemical properties, nutrient cycling and energy fluxes [23,24,25,26,27].

Consequently, ants alter the ecological filters that shape above- and belowground interactions by creating environments that constrain the species pool by promoting a set of biological traits of organisms that facilitate mutualistic associations with ants, thus supporting species coexistence [28,29,30]. For example, the engineering activity of ants can increase microbial processes and interactions, which in turn accelerate microbial exchange pools [31,32,33]. Ant soil engineering can also alter the soil seed bank [34,35]. Nevertheless, there are few studies investigating the effect of different ant species on belowground communities such as soil microarthropods, and most of the existing studies investigated wood ant (*Formica rufa* group) nests, which consist of aboveground mounds built with needles, leaves, twigs and bark material [36,37,38,39]. However, although some trends may be shared, the diversity and distribution patterns of soil microarthropods in the abovementioned mounds may differ significantly from those of microarthropods inhabiting mounds or small hills of granular soil built by soil-nesting ant species. Despite this, the latter have hardly been studied [40,41]. Consequently, little is known about the impact of soil-nesting ant species on the community structure and diversity patterns of soil microarthropod taxa associated with different trophic groups, such as detritivores and predators.

The contribution of ants to improving soil fertility is important for the maintenance of local biodiversity, particularly in highly disturbed habitats such as urban environments, where the soil’s suitability for plant growth may be compromised. In urban ecosystems, anthropogenic disturbances—including polluted precipitation, hydrological drought, changes in resource availability, and conversion to different land uses—result in altered soil conditions (see [42,43,44]). Nevertheless, urban green spaces that resemble natural plant communities (e.g., [45]) are commonly colonised by ants [46,47,48].

In this study, we conducted a survey of a multi-taxon belowground community of soil microarthropods—Collembola, Mesostigmata, Oribatida and Actinedida—occurring both in mounds and in the adjacent soil habitats of the two soil-nesting ant species *Lasius niger* (Linnaeus, 1758) and *Lasius flavus* (Fabricius, 1782) in urban areas. These two soil-nesting ant species differ greatly in their nesting and foraging habits. *L. niger* is a semi-predatory species with mostly epigean foraging habits (although it feeds in all biocoenotic layers), and often nests under stones [47]. *L. flavus* is a more cryptic species with hypogean foraging habits, feeding mostly on honeydew exuded by root aphids, and its mound nests are often overgrown with mosses, herbs and grasses [47]. In this study, we hypothesised that due to their different ecology, these two soil-nesting ant species differed in the type of soil disturbance caused by their activity. We further inferred that this would result in differences in the structure and composition of the soil microarthropod communities inhabiting their mounds and the surrounding areas. Our study aimed to identify gaps in the knowledge on biotic filtering in relation to ant–soil–microarthropod interactions, as well as to provide new insights into the influence of ant activity on the soil microarthropod community. Specifically, the main objectives of this study were to assess the effects of soil disturbance by soil-nesting ant species on the density and distribution of higher soil microarthropod taxa, and to evaluate the specific engineering effects of *L. flavus* and *L. niger* on the diversity patterns and structures of soil microarthropod assemblages with different trophic positions: detritivores and predators, represented by the two taxa Collembola and Mesostigmata (which are mainly free-living predatory species in soil litter habitats), respectively.

## 2. Materials and Methods

### 2.1. Study Sites

The study was conducted in semi-natural grasslands of open green spaces in the Warsaw agglomeration, where the soil-nesting ants *L. niger* and *L. flavus* are the most abundant ant species [47]. The three sites were located on frequently mowed (5 times a year) green lawns whose vegetation resembled pasture communities (Cynosurion alliance) or wet meadows (Arrhenatherion alliance) belonging to the order Arrhenatheretalia (Figure 1). Each site was geographically referenced using ArcView 9.3. The soils were classified as Cambisols and Luvisols formed from fluvioglacial sands [49].

### 2.2. Soil Microarthropod Sampling and Processing

To cover site variability, samples were taken from six mounds with living ant colonies: three from *L. niger* mounds and three from *L. flavus* mounds. A soil sample (5 cm in diameter × 10 cm deep) was taken from the centre of each mound. In addition, a control sample was taken 1 m from each mound in a grassland area with no ant-nesting activity. When selecting control sites, we considered the impact of ants on soil structure, growth of surrounding vegetation and root density in order to identify locations that were clearly unaffected by them. In total, 72 soil samples from ant mounds (36 from *L. niger* and *L. flavus*) and 36 soil samples from grasslands (control microhabitat) were collected twice during the growing season (June and September 2015).

Soil microarthropods from the mounds and adjacent control microhabitats were extracted in a modified MacFadyen high gradient apparatus for a maximum of 10 days. The material was then identified to the higher taxonomic levels of soil microarthropods (Collembola, Mesostigmata, Oribatida and Actinedida). Subsequently, the composition and structure of two assemblages of soil microarthropods, Collembola and Mesostigmata, representing different feeding habits—detritivores and predators, respectively—were assessed for the *L. niger* and *L. flavus* mounds.

Identification of Collembola species was performed using recent comprehensive monographs [50,51,52,53,54,55,56,57], following the nomenclature of [58]. The studied material was deposited in the Museum and Institute of Zoology, PAS (Warsaw, Poland).

To identify the species of Mesostigmata (Acari), both semi-permanent (using lactic acid) and permanent microslides (using Hoyer’s medium) were prepared. All the mesostigmatic mites were examined with a light microscope (Zeiss Axioskop 2, Carl Zeiss AG, Jena, Germany) and identified with reference to taxonomic literature (e.g., [59,60,61]).

### 2.3. Abiotic Factors

To characterise the microclimatic and soil conditions, we assessed soil moisture (soil volumetric water content, VWC), soil temperature (T_soil_) and electrical conductivity (ECe) using automated ProCheck sensors (Decagon Devices, Pullman, WA, USA). Measurements were taken from the central part of all ant mounds and control microhabitats in parallel with soil microarthropod sampling. Soil pH and total carbon, nitrogen, potassium and phosphorus content (hereafter referred to as major elements, quantified in g kg^−1^) were determined in the laboratory from the soil material collected in September 2015. Soil pH was measured using a 1:2.5 soil/water suspension by the potentiometric method. Total carbon (TC) and total nitrogen (TN) concentrations were quantified by the total combustion method using a LECO CNS 2000 automatic analyser (LECO Corporation: St. Joseph, MI, USA). The concentrations of total phosphorus (TP) and potassium (TK) were analysed by plasma–optical emission spectrometry after digestion of the samples in a mixture of nitric and perchloric acids [62], and quantified in g kg^−1^. The C/N nutrient ratio was calculated from these data.

### 2.4. Data Analysis

Differences in soil physicochemical properties and the abundance of soil microarthropod taxa (Collembola, Oribatida, Mesostigmata and Actinedida) between soil from mounds of *L. flavus* and *L. niger* and from control microhabitats (without ant-nesting activity), as well as between *L. niger* and *L. flavus* mounds, were compared using a two-sided Mann–Whitney U test. Differences in basic diversity parameters, such as species richness (S) and Shannon’s diversity index (H′) of Collembola and Mesostigmata assemblages between *L. niger* and *L. flavus* ant mounds were also compared using a two-sided Mann–Whitney U test. Principal component analysis (PCA) was then used to test whether the ants’ engineering activity associated with changes in soil properties was able to distinguish *Lasius* spp. mound microhabitats from control soils, as well as showing microhabitat differences between *L. niger* and *L. flavus* mounds. The PCA method, through the load vectors, allows for the assessment of soil physicochemical parameters that differentiate the examined microhabitats, and through the score vectors shows the range of variability in relation to the individual samples To determine the significance of differences between *Lasius* spp. mound and control microhabitats, and between *L. niger* and *L. flavus* mound microhabitats in the ordination space, the PCA scores of the first and second axes were used as a new synthetic dependent variables in the Mann–Whitney U test.

The effect of microhabitat type (M; ant mounds and control soil), season (S; spring, autumn) and site location (Sl; site 1, site 2, site 3) on the variation in soil microarthropod taxa (Collembola, Oribatida, Mesostigmata and Actinedida) was assessed by redundancy analysis (total RDA), since the value of gradient length calculated in the detrended correspondence analysis (DCA) for the first axis was shorter than 3.0 standard deviations (SD). Furthermore, the effect of species-specific ant mounds (AM) of *L. flavus* (LF) and *L. niger* (LN), season and site location on the variation in Collembola and Mesostigmata assemblages was tested using canonical correspondence analysis (total CCA), as the gradient length exceeded 3 SD (see [63]).

Variation partitioning was performed using partial models, including pRDA-(partial redundancy analysis) and pCCA (partial canonical correspondence analysis)-based models, to explain the unique (conditional) and shared effects of the investigated factors [64]. The aim of these models was to reflect the relative importance of microhabitat type, season and site location as a group of predictors of variation in soil microarthropod taxa and to represent the combined effect of ant mounds, season and site location on the variation in Collembola and Mesostigmata assemblages’ composition.

An interactive forward selection procedure was then used to assess the statistical significance of each individual explanatory variable of the factors studied—microhabitat type (M-ant mound, soil-control), season and site location—to rank the explanatory variables by their importance in determining the significance of their effect on variation in soil microarthropod taxa. The forward selection method was also used in a stepwise regression procedure to predict the significance of each variable for factors such as ant mound type, season and site location on variation in Collembola and Mesostigmata assemblages’ composition. In the model, the adjusted variation was applied using the number of degrees of freedom. A forward selection procedure (interactive) was used with 999 permutations, and the *p*-value was adjusted by the false discovery rate. The significance of the models was estimated using the unconstrained Monte Carlo permutation test.

We performed the multivariate analysis using the data matrix species and/or soil microarthropod taxa × ant mound collected at the site during the same sampling period, and then standardised to square metres. Before the multivariate analyses, data were log(x + 1)-transformed to meet assumptions of normality and homogeneity of variance. The significance level for all analyses was set at α = 0.05. Calculations were performed using the software packages Statistica 10.0 and Canoco 5.0.

## 3. Results

### 3.1. Soil Properties

The descriptive statistics of the recorded soil properties for the *Lasius flavus* mounds, *Lasius niger* mounds, the combined ant mounds (mounds of *L. niger* and *L. flavus* considered together, hereafter referred to as *Lasius* spp. and/or LF + LN), and the control microhabitat soils (C) are presented in Table 1. The two-tailed Mann–Whitney U test showed that the ant mounds (LF + LN) had significantly lower soil moisture (U = 466.5, z = −2.03, N = 36, *p* = 0.040) and higher total potassium (TK) (U = 421.0, z = 2.77, N = 18, *p* = 0.006) than the surrounding control grassland soils. However, there were no significant differences in the mean ranks of the other soil characteristics measured between *L. flavus* (LF) and *L. niger* (LN) mounds.

The results of the PCA ordination confirmed the differences in soil properties between ant mounds (LF + LN) and control microhabitats (C) without ant-nesting activity, with significantly higher factor loadings for the first PC axis (two-tailed Mann–Whitney U test for sum range of soil properties: *p* < 0.05; *n* = 36 in all cases; Figure 2A). PCA axis 1 accounted for 73.9% of the total variance in soil properties, with variables such as total nitrogen (TN), carbon (TC) and phosphorus (TP) contents having the largest positive loadings; in contrast, soil temperature, followed by soil pH, had the largest negative loadings. PCA axis 2 accounted for 27.0% of the total variance and had the largest positive loading on potassium (TK) and a negative loading on soil moisture (Figure 2B, Appendix A).

The results of further PCA ordination revealed differences in soil properties between *L. flavus* and *L. niger* mounds. Significantly larger factor loadings were found for the first PC axis (two-tailed Mann–Whitney U test for sum range of soil properties: *p* < 0.05; *n* = 18 in all cases; Figure 3A). PCA axis 1 accounted for 78.1% of the total variance in soil properties. The highest negative factor loading was found for soil temperature, while the highest positive factor loading was found for TC, TN and TP content. PCA axis 2 accounted for 12.7% of the total variance and had the highest factor loadings positively related to EC and negatively related to TK and moisture (Figure 3B, Appendix A). The soils of *L. niger* mounds were slightly less warm and moist and more fertile (characterised by higher TN, TC, TP and TK contents) than *L. flavus* mounds.

### 3.2. Soil Microarthropod Taxonomic Composition and Abundance Patterns

Descriptive statistics of the recorded abundances of soil microarthropod taxa from the *L. flavus* and *L. niger* mounds and the control (surrounding grassland) soil are presented in Table 2. A two-tailed Mann–Whitney U test showed that the abundance of soil microarthropods with generalist feeding habits—i.e., Collembola and Oribatida (classified as detritivores)—was significantly reduced within the ant mounds (LF + LN) compared to the control microhabitats (two-tailed Mann–Whitney U test for Collembola: *p* = 0.0002; Oribatida: *p* = 0.0009; *n* = 36 in all cases). Conversely, the abundance of soil microarthropods characterised by specialised feeding habits—i.e., Mesostigmata and Actinedida (classified as predators)—remained constant between the ant mounds and control microhabitats. In contrast, the abundance of soil microarthropod taxa in *L. flavus* and *L. niger* mounds did not differ significantly (two-tailed Mann–Whitney U test *p* > 0.05, *n* = 18) (Table 2).

The results of the total RDA model confirmed that the study factors (the soil microhabitat—i.e., nested ant mound/non-nested control grassland soil—as well as the site and season) accounted for 12.8% of the variation (adjusted explained variation 7.6%). The results of this model explained a significant amount of variation in soil microarthropod components (λ-trace = 0.1284, F-ratio = 2.5, *p*-value = 0.008). This analysis also showed that the presence of soil-nesting mounds of both ant species had a negative influence on the distribution pattern of soil microarthropod taxa (Figure 4).

Variation decomposition with the partial RDA model revealed that soil microhabitat (M) and season (S) had the highest unique effects within the model, accounting for 5.2 and 4.2% of the adjusted variation, respectively, while the contribution of the site factor to the total variation was negative (−1.5%) and non-significant. Furthermore, the joint overlap of all three factors was relatively small (*p* < 0.1%; Table 3).

Further analysis using partial model RDA with interactive forward selection identified the significant effects of soil-nesting ant mound, control site and season (spring and autumn) as single variables on the Variation in Soil microarthropod communities (Table 4).

The results of subsequent RDA and partial RDA analyses showed that the abundance of soil microarthropods did not respond to environmental changes (soil disturbance) in *L. niger* and *L. flavus* ant mounds (Appendix A).

### 3.3. Diversity Pattern and Structure of Collembola Assemblages

A total of 25 Collembola species were recorded in the ant mounds of *L. niger* and *L. flavus* (Appendix A). The majority of species recorded were soil-dwelling, while the myrmecophilous species were represented by only two species, *Cyphoderus albinus* and *C. bidenticulatus*. The effect of *L. flavus* and *L. niger* ant mounds on the mean ranks of collembolan abundance (A) was not significant (Table 2), species richness (S) and Shannon’s diversity index (H′) were also not statistically significant (two-tailed Mann–Whitney U test (N = 18, *p* > 0.05 in all cases).

The total CCA analysis showed that all the investigated Factors—LN Vs. LF ant mounds, season and site—accounted for 17.4% of the variation (adjusted explained variation 3.6%) in the species dataset (λ-trace = 1.98, F-ratio = 1.30, *p*-value = 0.02). Variation decomposition (Table 5) showed that season had the greatest unique effect, accounting for 3.1% of the adjusted variation, while ant mounds and site were not significant and explained less of the variation (0.7% and 0.6%, respectively). The results of the partial CCA with site as a covariate showed further significant differences in Collembola assemblage composition between ant mounds (λ-trace = 0.702, F-ratio = 1.481, *p*-value = 0.008). For example, during the autumn season, myrmecophilous collembolan species such as *C. bidenticulatus* and *C. albinus* were observed to be associated with *L. flavus* mounds (Appendix A).

### 3.4. Diversity Pattern and Structure of Mesostigmata Assemblages

A total of 21 Mesostigmata species were recorded in the ant mounds of *L. niger* and *L. flavus* (Appendix A). Most of the species recorded were soil-dwelling. The effect of *L. flavus* and *L. niger* ant mounds on the mean ranks of Mesostigmata abundance (A) was not significant (Table 2); the species richness (S) and Shannon’s diversity index (H′) were also not statistically significant (two-tailed Mann–Whitney U test (N = 18, *p* > 0.05 in all cases).

The total CCA analysis showed that all predictors included in the model accounted for 15.9% of the variation (adjusted explained variation 0.6%), but the model was not significant (λ-trace = 1.475, F-ratio = 1.0, *p*-value = 0.372). Further variation decomposition and partial CCA analysis with site as a covariate were not significant (Table 6, Appendix A).

## 4. Discussion

Our results, as expected, showed that ants can shape soil microarthropod taxa. These results are consistent with other studies showing that ant-induced soil disturbance can cause changes in belowground communities [19,28,36,39].

The effect of ant ecological pressure on the community structure of soil microarthropods in both types of microhabitats studied (soil-nesting ant mounds vs. control soil) was confirmed by the differences found in the abundance of soil microarthropods at the coarse taxonomic level (Table 2). However, our study showed that this effect varied according to the position of the soil microarthropod taxon in the soil food web. In particular, ant pressure had a negative effect on the abundance of taxa with decomposition/detritivore functions in soil (i.e., Collembola and Oribatida) and a neutral effect on the abundance of taxa considered to be top predators in the soil mesofauna food web, i.e., Mesostigmata [12] (Table 2). This result confirms the importance of edaphic microarthropod food specialisation as a key factor influencing community structure and function [65], even in microhabitats with ant-induced soil disturbance.

Furthermore, we found significant variability in soil microarthropod communities between the studied microhabitats—with and without ant-nesting activity—using the RDA and pRDA models (Table 3 and Table 4, Figure 4), as well as the significant effect of seasonal variation on these communities. These results suggest that the selective filter shaped by ants through which soil microarthropod taxa pass—in addition to exposure to chronic predation—is related not only to disturbance of the soil environment within mounds as a result of ant-nesting activity, but also to variation in microhabitat conditions between mounds and control soils over time.

Several authors have indicated that ants, through their nesting activity, directly and indirectly alter both biotic and abiotic properties of the soil environment within the mound [1,23,24,66,67], including soil physicochemical properties or resource availability [27,28,66,67,68]. Ant-digging activity also induces structural disturbances associated with fine-scale landscape-level changes, which enhance basic physical processes such as water infiltration and nutrient redistribution [69]. Our results showed that the negative response of soil microarthropod communities to soil disturbance by soil-nesting ants was mainly associated with significantly increased soil dryness and significantly higher contents of soil-fertility-enhancing macronutrients in mound soils compared to control soils without ant-nesting activity (Table 1 and Appendix A, Figure 2A,B).

Soil microarthropods are highly sensitive to environmental changes caused by drought and soil warming [70,71,72,73,74,75]. Disturbances in soil physicochemical properties, such as soil drought/temperature and/or soil fertility, can strongly influence their community structure through bottom-up effects [71,75,76,77]. In addition, changes in hydration status, which is a key descriptor of soil physical structure and an important driver of trophic interactions in belowground communities [77], can locally facilitate predator–prey interactions, leading to a sharp decrease in the abundance of prey communities.

Conversely, changes in microhabitat conditions in the ant nests of *Lasius* spp. due to their activity, particularly those associated with increases in soil fertility, did not affect the abundance of Mesostigmata (Table 2). Mesostigmata is a taxon considered to be an important predator in the food web of soil mesofauna [78]. This result contradicts the study by [76], who found an increased proportion of predatory soil microarthropods in temperate grassland communities with improvements in soil fertility. The present results are also inconsistent with the study of [39], who reported a negative effect of red ants (*Formica polyctena*) on microarthropod taxa in a semi-arid pine forest. These discrepancies may be due to different degrees of soil pH disturbances influenced by the specific ant species, which in turn affects soil-associated biodiversity. Red ants have been shown to alter soil pH, shifting acidic and neutral soils to more alkaline soils, and strongly alkaline soils to more acidic soils [79]. However, our study found no significant differences in soil pH between areas with *Lasius* spp. and areas without ant-nesting activity. Furthermore, the present study differs from that of [39] in terms of the composition of the organisms studied, as the present study included soil consumers of the same size class and within the same microhabitat network [14,78]. Despite the absence of differences in the abundance of Mesostigmata mites in the *Lasius* spp. vs. control microhabitats studied, these organisms may still exert a cascading effect on mesofaunal detritivore taxa. This finding suggests that the density of soil microarthropods in ant nests is not predominantly controlled by bottom-up forces, but rather by top-down regulation.

Nevertheless, it is noteworthy that the nest mounds of *L. niger* and *L. flavus* did not show species-specific differences in the abundance and occurrence of higher soil microarthropod taxa (Table 2) although both species differ in their ecological characteristics [47]. This scenario may be related to their sympatric coexistence and nesting in disturbed habitats, such as urban green spaces. The differences observed in the impact of these two ant species on soil properties in the urban environment: *L. niger* mounds were slightly less warm and moist and more fertile than *L. flavus* mounds (Table 1 and Appendix A, Figure 3B) did not cause any notable changes in soil microarthropod communities. However, the published results show that moisture and temperature within ant mounds are important for soil microarthropods [38].

### 4.1. Effect of L. flavus and L. niger on Collembola Assemblage Composition, Abundance and Distribution Patterns

The present study showed that microhabitat variation between *L. flavus* and *L. niger* mounds did not significantly affect differences in Collembola abundance, species richness or Shannon diversity index values. However, a more detailed analysis of the Collembola assemblages inhabiting these ant nests suggests that seasonal variation may play an important role in shaping their assemblages. Analysis of the species composition of the Collembola assemblages found in the nests of the *L. flavus* and *L. niger* indicates that most of the assemblages are species that prefer (inhabit) the litter and soil layers. A similar pattern was observed in the Collembola assemblages found in the anthills of *Formica* s. str. nests [37]. However, it is noteworthy that the anthills of *L. flavus* and *L. niger* also host two additional myrmecophilous species, *Cyphoderus albinus* and *C. bidenticulatus*. Despite the well-documented association of the genus *Cyphoderus* (Nicolet, 1842) with the nests of social insects—including termites, bees and ants (see [58,80])—*C. bidenticulatus* was found only in *L. flavus* nesting mounds (Appendix A), suggesting that the microhabitat conditions created by this ant species may be particularly conducive to its presence.

The variance partitioning results also showed that the variation in Collembola assemblages in *L. niger* and *L. flavus* mounds was significantly influenced by seasonal changes (see Table 5). The present study did not assess the extent to which seasonal changes in habitat conditions occur in the mounds of both ant species. However, it is recognised that seasonal changes in grassland ecosystems can induce a response in soil microarthropod assemblages, including those of Collembola, and that these changes are mainly related to variations in temperature and precipitation [69,72]. Nevertheless, it is noteworthy that despite the lack of significant variability in Collembola assemblages associated with the unique microhabitats of *L. niger* and *L. flavus*, the drier microhabitat conditions found in *L. niger* nest mounds (compared to *L. flavus* mounds) favoured the occurrence of xerophilic Collembola species such as *Isotomodes productus* and *Hemisotoma thermophila*. We also found higher concentrations of key soil-fertility components (C, N, K and P) in *L. niger* mounds, creating a microhabitat favourable for the presence of species associated with habitats rich in decomposing organic matter, such as *Proisotoma minima* (see [54]). However, the distribution pattern of Collembola species observed in *L. flavus* and *L. niger* mounds may be impacted by variables outside the scope of this study. For example, it is important to note that the species identified in this study have been selected for by urban environmental conditions, and that the colonisation of soil-nesting ant mounds by Collembola assemblages may be affected by both anthropogenic and natural disturbances.

### 4.2. Effect of L. flavus and L. niger on Mesostigmata Assemblage Composition, Abundance and Distribution Patterns

The present study showed that microhabitat variation between *L. flavus* and *L. niger* mounds had a significant influence on the abundance of Mesostigmata mite assemblages, but it did not show a discernible effect on changes in mite diversity (see Appendix A). However, it is noteworthy that the studied ant mounds contained almost half as many Mesostigmata mite species as the 53 and 44 species previously identified in *L. flavus* and *L. niger* nests in Poland [40,81]. A review of over 40 years of research in Poland on Mesostigmata mites in *L. flavus* and *L. niger* nests [82,83,84] shows high species diversity, with each study typically including a different list of species. This diversity is probably due to the function of the soil as a microhabitat where nests are built, often in humus or mineral layers, supporting numerous mite species. Consequently, the lower diversity of Mesostigmata mites observed in our study may be due to the effects of urban disturbance on these communities (e.g., [85]). Changes in the microhabitat characteristics of the ant mounds of the studied *Lasius* species are also reflected in the presence of Mesostigmata species frequently found in the mineral soil layer (e.g., *Rhodacarellus silesiacus*, *Rhodacarus mandibularis*) and in the upper litter layer (e.g., *Gamasellodes bicolor*, *Veigaia nemorensis*). In contrast to the myrmecophilous Mesostigmata found in the nest mounds of red-headed ants (*Formica rufa* and *F. polyctena*) [86], only *Cosmolaelaps vacua* and *Oplitis conspicua*—two soil-dwelling Mesostigmata species often associated with ant nests—were found in the nest mounds of *L. niger* and *L. flavus* [81].

## 5. Conclusions

Our research has identified the role of ecological filters in the response of soil microarthropod taxa to ant engineering activity. As we expected, soil disturbance caused by ant engineering activity can influence the response of soil microarthropod taxa, including their abundance and distribution patterns. Digging and mound building by the two ant species studied (*L. niger* and *L. flavus*) not only increased the local heterogeneity of grassland communities in urban green spaces, but also induced soil disturbance that affected the variability of soil microarthropod assemblages. However, species-specific disturbance and the associated change in microhabitat conditions in the mounds of the two ant species studied did not significantly affect the variation in diversity patterns and species composition of the Collembola and Mesostigmata assemblages. In contrast, the results of the present study suggest that seasonal variation in resource availability in mounds and interspecific interactions may be important factors, especially in the assembly of detritivorous—but not predatory—guilds of soil microarthropods.

However, it should be noted that while urban green spaces tend to resemble natural plant communities, anthropogenic disturbances (e.g., pollutant deposition, hydrological droughts, changes in resource availability, and conversion to different land uses) also strongly modify soil (see [44]), including that of the myrmecosphere. Consequently, the response exhibited by the local soil microarthropod community to ant engineering activity in urban areas may be either universal or unique.

## Figures and Tables

**Figure 1 insects-16-01058-f001:**
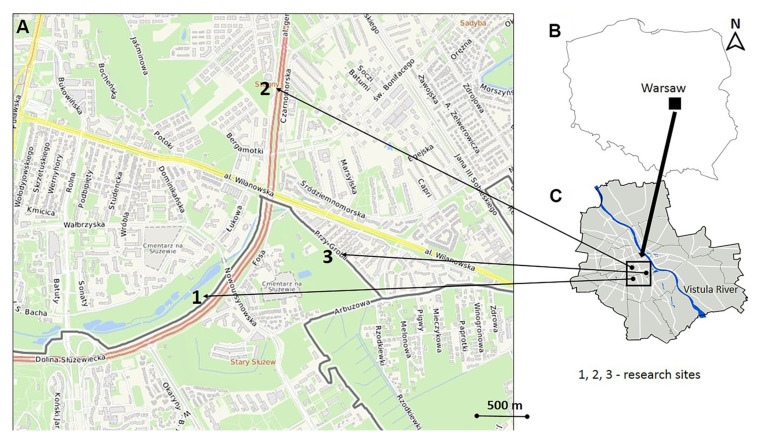
Location of the study area and distribution of the examined semi-natural grassland sites in the urban green space. (**A**)—map of the study sites in the open green areas in the Warsaw agglomeration, (**B**)—location of the study area in Poland, (**C**)—location of the study area in the Warsaw agglomeration. Site 1—52°10′10″ N 21°02′03″ E; Site 2—52°10′49″ N 21°02′35″ E; Site 3—52°10′12″ N 21°02′58″ E.

**Figure 2 insects-16-01058-f002:**
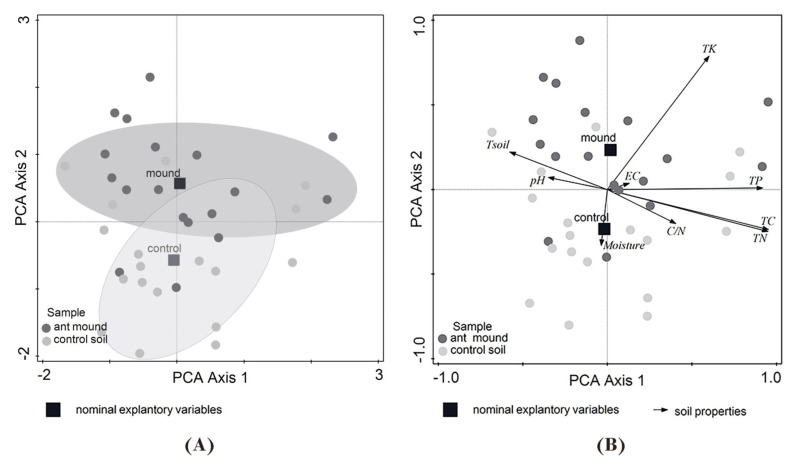
The PCA (**A**) ordination biplot (PC 1 and PC 2) and (**B**) correlation biplot (PC 1 and PC 2) of the soil-nesting *Lasius flavus* + *Lasius niger* mounds and non-nested (control) plots, with the microhabitat type (ant-mound and control) as the supplementary variables. The model was calculated using raw data of soil characteristics and interspecies correlation scaling, with the species score divided by the standard deviation (SD) and centred by species, not standardised by sample.

**Figure 3 insects-16-01058-f003:**
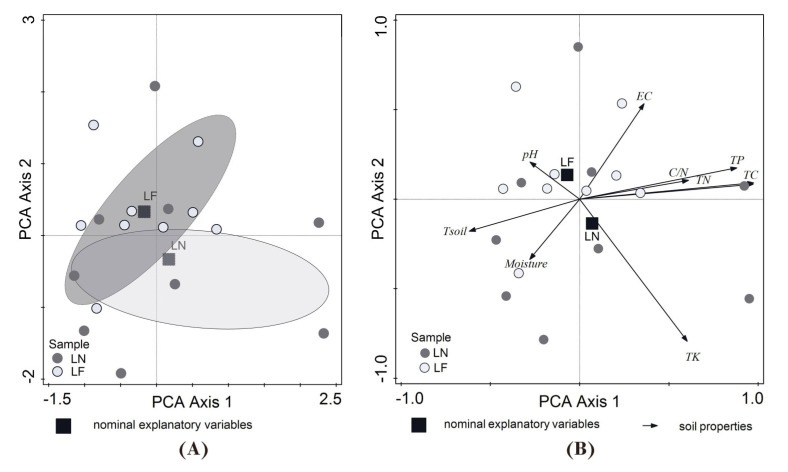
The PCA (**A**) ordination biplot (PC 1 and PC 2) and (**B**) correlation biplot (PC 1 and PC 2) of the soil-nesting *Lasius flavus* (LF) and *Lasius niger* (LN) mounds with the microhabitats of ant-mounds as the supplementary variables. The model was calculated using raw data of soil characteristics and interspecies correlation scaling, with the species score divided by the standard deviation (SD) and centred by species, not standardised by sample. Soil-characteristic abbreviations are given in Table 1.

**Figure 4 insects-16-01058-f004:**
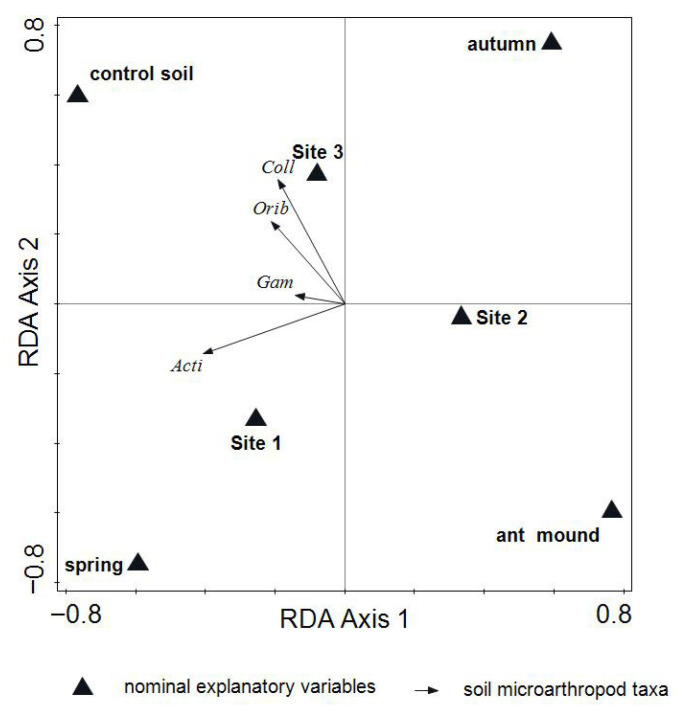
Distribution pattern of soil microarthropod taxa within soil-nesting ant mounds and non-nested (control) sites. RDA model calculated with log(x + 1) transformed data. Coll—Collembola; Orib—Oribatida; Mes—Mesostigmata; Acti—Actinedida.

**Table 1 insects-16-01058-t001:** Summary statistics (mean ± standard deviation SD) of soil characteristics from the ant mounds of *Lasius niger* (LN), *Lasius flavus* (LF), *Lasius flavus* (LF) + *Lasius niger* (LN), and control microhabitats (C) without ant-nesting activity in open green space in the Warsaw agglomeration. TK—total potassium content; TP—total phosphorus content; TN—total nitrogen content; TC—total carbon content; EC—electrical conductivity. Values presented in the table are means across replicates of each site for soil moisture, T_soil_ and EC (*n* = 36), and for TK, TP, TN, TC and C/N ratio (*n* = 18).

Soil Properties	LN Mound		LF Mound		LN + LF		Control (C)		*p*-Value ^a^	*p*-Value ^b^
	Mean	SD	Mean	SD	Mean	SD	Mean	SD		
pH_H20_	6.86	0.45	6.87	0.38	6.87	0.40	6.77	0.49	0.863	0.658
TK (g kg^−1^)	1.45	0.49	1.16	0.19	1.31	0.39	0.99	0.32	0.222	**0.006**
TP (g kg^−1^)	0.84	0.37	0.82	0.22	0.83	0.30	0.79	0.32	0.796	0.517
TN (g kg^−1^)	6.21	2.67	5.37	1.38	5.79	2.10	6.20	2.03	0.863	0.420
TC (g kg^−1^)	72.27	34.94	58.96	14.71	65.61	26.89	70.15	23.31	0.863	0.420
C/N ratio	11.44	0.75	11.00	0.30	11.22	0.60	11.30	0.35	0.258	0.137
Moisture (%)	12.20	5.05	14.16	4.61	13.18	4.86	15.91	5.36	0.252	**0.040**
T_soil_ (°C)	25.90	4.79	27.72	4.56	26.81	4.70	26.79	4.65	0.161	0.978
EC (µS cm^−1^)	0.03	0.04	0.03	0.02	0.03	0.03	0.02	0.02	0.888	0.827

Significant *p*-values are in bold (*p* < 0.05) tested by a two-tailed Mann–Whitney U test (d.f = 1) *p*-value ^a^ corresponds to differences between LN and LF; *p*-value ^b^ corresponds to differences between ant mounds (LN + LF) and control microhabitats (C).

**Table 2 insects-16-01058-t002:** Abundance of soil microarthropod taxa in *Lasius flavus* (LF), *Lasius niger* (LN), and *Lasius flavus* + *Lasius niger* (LF + LN) ant mounds, and in control microhabitats without ant-nesting activity (C) in open green space in the Warsaw agglomeration. Values are means ± standard deviation (SD) of 18 replicates of taxon abundance in LF and LN ant mounds, and 36 replicates of taxon abundance in LF + LN ant mounds and control microhabitats (C); D—density of soil microarthropod taxa (ind.m^−2^).

Microarthropod Taxon	LN Mound		LF Mound		LN + LF		Control (C)		*p*-Value ^a^	*p*-Value ^b^
	Mean	SD	Mean	SD	Mean	SD	Mean	SD		
Collembola	4444.44	3289.42	4361.11	5315.92	4402.79	4356.96	9638.89	8072.54	0.365	**<0.001**
Mesostigmata	4138.89	4186.13	1583.33	2088.13	2861.11	3508.38	2972.22	3200.32	0.076	0.505
Oribatida	10,111.11	11,245.33	3638.89	3109.78	6875.00	8768.76	18,013.89	21,163.14	0.062	**<0.001**
Actinedida	2527.78	4974.86	1222.22	2157.22	1875.00	3836.62	1888.89	1848.21	0.584	0.086

Significant *p*-values are in bold (*p* < 0.05) tested by a two-tailed Mann–Whitney U test (d.f = 1), *p*-value ^a^ corresponds to differences between LN and LF; *p*-value ^b^—corresponds to differences between ant mounds (LN + LF) and control microhabitats (C).

**Table 3 insects-16-01058-t003:** Variance partitioning among soil microhabitats (M: ant mounds + control), site (Sl) and season (S) on variation in soil microarthropod communities (based on higher taxonomic ranks) from soils in green spaces in Warsaw. Conditional effect performed by RDA and partial RDA model. DF—degree of freedom; mean square—denominator of F-statistic, pseudo-F—F statistic, *p*-value—significance level of the effect tested by Monte Carlo permutation test, RDA model calculated with log(x + 1) transformed data.

Model Fraction	Explained Variation (%)	Contribution to the Total Variation (%)	DF	Mean Square	Pseudo-F	*p*-Value
Soil microhabitat M (ant mound + control soil)	68.5	5.2	1	0.063	4.9	**0.040**
Site Sl	−19.2	−1.5	2	0.006	0.5	0.884
Season S	55.4	4.2	1	0.054	4.1	**0.004**
Overlap of M + Sl	−1.7	−0.1				
Overlap of M + S	−1.3	−0.1				
Overlap of Sl + S	−1.8	−0.1				
Joint overlap of M + Sl + S	<0.1	<0.1				
Total explained	100	7.6	4	0.032		
All variation	1	100	71			

Significant *p*-values are indicated in bold. Variations account using the adjusted R2 approach.

**Table 4 insects-16-01058-t004:** Results of interactive forward selection in RDA and ranking of examined factors on variation in the soil microarthropod communities. Model calculated with log(x + 1) transformed data; pseudo-F—F statistic; *p*-value—significance level of the effect tested by Monte Carlo permutation test with 999 permutations; *p*-value (adj.)—correction with false discovery rate.

Variable	Contribution (%)	Pseudo-F	*p*-Value	*p*-Value (adj)
Soil microhabitat: ant mound	49.2	4.7	**0.008**	0.056
Soil microhabitat: control	49.2	4.7	**0.002**	**0.014**
Season: spring	41.7	4.2	**0.008**	0.056
Season: autumn	41.7	4.2	**0.007**	**0.028**
Site: site 2	4.9	0.5	0.716	0.784
Site: site 1	4.2	0.4	0.780	0.806

Significant *p*-values are indicated in bold. Variations account using the adjusted R2 approach.

**Table 5 insects-16-01058-t005:** Variance partitioning among ant mounds (AM: *Lasius niger* and *Lasius flavus*), site (Sl) and season (S) on variation in Collembola assemblages from soils in green spaces in Warsaw. Conditional effect performed by CCA and partial CCA model. DF—degree of freedom; mean square—denominator of F-statistic; pseudo-F—F statistic; *p*-value—significance level of the effect tested by Monte Carlo permutation test; CCA model calculated with log(x + 1) transformed data.

Model Fraction	Explained Variation (%)	Contribution to the Total Variation (%)	DF	Mean Square	Pseudo-F	*p*-Value
Ant mound AM	20.2	0.7	1	0.281	1.2	0.200
Site Sl	15.9	0.6	2	0.255	1.1	0.334
Season S	86.6	3.1	1	0.430	1.8	**0.006**
Overlap of AM + Sl	−8.4	−0.3				
Overlap of AM + S	−8.4	−0.3				
Overlap of Sl + S	−11.5	−0.4				
Joint overlap of AM + Sl + S	5.7	0.2				
Total explained	100	3.6	4	0.299		

Significant *p*-values are indicated in bold. Variations account using the adjusted R2 approach.

**Table 6 insects-16-01058-t006:** Variance partitioning among ant mounds (AM), site (Sl) and season (S) on variation in Mesostigmata assemblages from soils in green space in Warsaw. Conditional effect performed by CCA and partial CCA model. DF—degree of freedom; mean square—denominator of F-statistic; pseudo-F—F statistic; *p*-value—significance level of the effect tested by Monte Carlo permutation test; CCA model calculated with log(x + 1) transformed data.

Model Fraction	Explained Variation (%)	Contribution to the Total Variation (%)	DF	Mean Square	Pseudo-F	*p*-Value
Ant mound AM	20.6	0.1	1	0.365	1.0	0.428
Site Sl	138.8	0.8	1	0.318	1.1	0.282
Season S	−78.9	−0.5	2	0.390	0.9	0.688
Overlap of AM + Sl	−2.8	−0.0				
Overlap of AM + S	6.7	<0.1				
Overlap of Sl + S	11.6	<0.1				
Joint overlap of AM + Sl + S	4.1	<0.1				
Total explained	100		4	0.369		
All variation		100	26			

Variations account using the adjusted R2 approach.

## Data Availability

The data used in this study are available through the link to https://doi.org/10.6084/m9.figshare.29716343.v2.

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
