# Peer review of "Impact of Biogenic Structures of the Soil-Nesting Ants Lasius niger and Lasius flavus on the Soil Microarthropod Community in Urban Green Spaces"

_insects, 2025, doi:10.3390/insects16101058_

Round 1

Reviewer 1 Report

Comments and Suggestions for Authors

The manuscript needs to be slightly improved in accordance with the comments and clarifications:

  • line 67-68. Please add a link to https://dx.doi.org/10.24189/ncr.2024.018
  • line 78-79. Please add a link to https://dx.doi.org/10.24189/ncr.2024.013 
  • Specify the exact purpose and objectives of your research
  • The authors carried out the identification of taxa. However, there are no species lists in the manuscript and it is difficult to assess the species diversity. I suggest making lists of species in a separate application.
  • line 402-411. This is unnecessary information. Add this information to the introduction.
  • The link https://www.mdpi.com/article/doi/s1 inactive. It is difficult to evaluate some of the figures and tables.

Author Response

Thank you very much for taking the time to review this manuscript. Please find the detailed responses below and the track changes in the re-submitted files

 Response to comments and suggestions

Comments 1:

The manuscript needs to be slightly improved in accordance with the comments and clarifications:

Response 1:

Thank you very much for taking the time to review this manuscript. Please find the detailed responses below and the track changes in the re-submitted files

Comments 2 and 3:

line 67-68. Please add a link to https://dx.doi.org/10.24189/ncr.2024.018

line 78-79. Please add a link to https://dx.doi.org/10.24189/ncr.2024.013 

Response 2 and 3

Thank you very much for your suggestions. However, we feel that the list of references in our work is already too long and would prefer not to extend it further.

Comments 4:

Specify the exact purpose and objectives of your research

Response 4

At the end of the Introduction chapter, we presented our hypothesis and the objectives of the study as follows (lines 132-145)

“In this study, we hypothesised that due to their different ecology, these two soil-nesting ant species differed in the type of soil disturbance caused by their activity. We further inferred that this would result in differences in the structure and composition of the soil microarthropod communities inhabiting their mounds and the surrounding areas. Our study aimed to identify gaps in the knowledge on biotic filtering in relation to ant–soil–microarthropod interactions, as well as to provide new insights into the influence of ant activity on the soil microarthropod community. Specifically, the main objectives of this study were to assess the effects of soil disturbance by soil-nesting ant species on the density and distribution of higher soil microarthropod taxa, and to evaluate the specific engineering effects of the two soil-nesting ant species L. flavus and L. niger on the diversity patterns and structures of soil microarthropod assemblages with different trophic”

Comments 5:

The authors carried out the identification of taxa. However, there are no species lists in the manuscript and it is difficult to assess the species diversity. I suggest making lists of species in a separate application

Response 5

Indeed, the data sets under consideration also include information regarding the species composition of Collembola and Mesostigmata in the nests of two ant species: L. niger and L. flavus. The file entitled 'manuscript-supplementary.pdf' contains data on species composition. The information on the species composition of Collembola and Mesostigmata has been relocated to the supplementary materials in order to avoid an excessive increase in the article's length. The aforementioned tables are cited in the text as 'Table with the extension S'. The PDF file containing supplementary tables and figures is re-attached

Comments 6

line 402-411. This is unnecessary information. Add this information to the introduction.

Response 6

Indeed, in the opening paragraph of the discussion, the primary theses of the work were briefly revisited.

However, we agree that this part of the discussion is redundant.

Line 402-411 – we corrected the part as follow:

One of the main objectives of the present study was to assess the impact of 402 soil-nesting ants’ biogenic structures on belowground communities, with a particular 403 focus on the soil microarthropod taxa Collembola, Oribatida, Mesostigmata and Ac-404 tinedida. To achieve this goal, the study covered three semi-natural grasslands in the 405 Warsaw metropolitan area, where the abundance and distribution of soil microarthro-406 pods were investigated in the mounds of two different species of soil-nesting 407 ants—Lasius flavus (LF) and Lasius niger (LN)—as well as in grassland soils with no 408 ant-nesting activity (control). We also investigated the species-specific effects of the bio-409 genic structures of these two ant species on the diversity and structure of soil microar-410 thropod assemblages at different trophic levels of the soil food web: decomposers (Col-411 lembola) and predators (Mesostigmata). Our results, A as expected, our results showed that ants can 412 shape soil microarthropods taxa. These results are consistent with other studies showing 413 that ant-induced soil disturbance can cause changes in belowground communities 414 [17,26,34,37].

Comments 7:

The link https://www.mdpi.com/article/doi/s1 inactive. It is difficult to evaluate some of the figures and tables.

Response 7

We re-attaching the pdf file entitled manuscript-supplementary. We hope that the link will be activated once the DOI number has been published

  1. Response to Comments on the Quality of English Language

Point 1: quality of language English – English does not require any improvements

Response 1: Thank you

  1. Additional clarifications

[We are re-attaching manuscript-supplementary pdf].

Reviewer 2 Report

Comments and Suggestions for Authors

I found an interesting study because the authors combine very detailed information on the study system. The fact that the two ant species have different biology is also well-planned. I have just a few comments regarding mainly material and methods. I like most of the organization of the Introduction and Discussion. Regarding text, I would avoid the use of ‘two ant species, L. flavus and L. niger’, using or ‘two ant species’ or ‘L. flavus and L. niger’. This would shorten the text in a few places.

M&M:

- not clear how the three sites cited on line 150 is connected with sampling cover (the 72 soil samples); further, it’s necessary to include the reason to test for site effects in analyses (for example, distance between site 1 and site 3, around 1,500m);

- line 165: I would like to see an explanation if 1m apart from the ant mound can be considered a control site (maybe citing source references or clarifying the known size underground occupied by the ant nests);

- there are a lot of analyses (rank tests, PCA, total RDA, total CCA, p-RDA, pCCA); please, provide further details about the need to apply rank tests on PCA axes; not sure what is being further extracted in testing axes values. If you are in search of a multivariate variable (PC1-PC2), this should be clear in the text; PCA figures are ok, they allow us to understand better differences (microhabitat).

Author Response

Response to Reviewer 2

I found an interesting study because the authors combine very detailed information on the study system. The fact that the two ant species have different biology is also well-planned. I have just a few comments regarding mainly material and methods. I like most of the organization of the Introduction and Discussion. Regarding text, I would avoid the use of ‘two ant species, L. flavus and L. niger’, using or ‘two ant species’ or ‘L. flavus and L. niger’. This would shorten the text in a few places.

Thank you very much for your positive opinion about our work. Please find the detailed responses below and the corresponding corrections highlighted in track changes in the re-submitted files. We have consistently introduces the proposed shortens into the text

Response to comments and suggestions

Comments 1:

not clear how the three sites cited on line 150 is connected with sampling cover (the 72 soil samples); further, it’s necessary to include the reason to test for site effects in analyses (for example, distance between site 1 and site 3, around 1,500m);

Response 1: In our analysis, we considered the impact of site location, since differences in habitat conditions between sites could affect the variability of soil microarthropod communities. In multivariate analysis, we examined the 'site effect' using dummy variables, which are a type of explanatory variable that is used to represent categorical data numerically. The following explanation was added to the data analysis:

“..To determine the significance of heterogeneity among the studied sites for the variability of soil microarthropod taxa, the effect of site location was considered in the study.'

We also corrected the fragment in line 150 as follows “In total, 72 soil samples from ant mounds (36 from L. niger and 36 from L. flavus) and 72 36 soil samples from grasslands (control microhabitat) were collected twice during the growing season (from June and to September 2015).

Comments 2:

line 165: I would like to see an explanation if 1m apart from the ant mound can be considered a control site (maybe citing source references or clarifying the known size underground occupied by the ant nests);

Response 2

When selecting control sites, we considered the impact of ants on soil structure, surrounding vegetation and roots density to identify a location clearly unaffected by them. Control soils, without ants activity, has a more stable and consolidate structure, less vigorous growth of vegetation and higher root/rhizome biomass. Our observations indicated that a distance of one meter from the ant mound could be considered an appropriate control in an urban environment. The differences in habitat conditions between the ant mounds and the control site, particularly with regard to nutrients, confirmed this hypothesis.

We have expanded the fragment as follows

“When selecting control sites, we considered the impact of ants on soil structure, growth of surrounding vegetation and roots density in order to identify locations that were clearly unaffected by them”.

Comments 3

there are a lot of analyses (rank tests, PCA, total RDA, total CCA, p-RDA, pCCA); please, provide further details about the need to apply rank tests on PCA axes; not sure what is being further extracted in testing axes values. If you are in search of a multivariate variable (PC1-PC2), this should be clear in the text; PCA figures are ok, they allow us to understand better differences (microhabitat).

Response 3

Thank you for your comment. We will correct the fragment as follows:

“..Principal component analysis (PCA) was then used to test whether the ants’ engineering activity associated with changes in soil properties was able to distinguish Lasius spp. mound microhabitats from control soils, as well as showing microhabitat differences between L. niger and L. flavus mounds.

The PCA method, through the load vectors, allows for the assessment of soil physicochemical parameters that differentiate the examined microhabitats, and through the score vectors shows the range of variability in relation to the individual samples. To determine the significance of differences between Lasius spp. mound and control microhabitats, and between L. niger and L. flavus mound microhabitats  in the ordination space, the PCA results scores of the first and second axes were used as a new synthetic dependent variables in the Mann–Whitney U test.”

In PCA analysis, two results are generated, loadings and score vectors. The loading vectors describe the direction of the principal components in relation to the original variables, whereas the score vectors describe the direction of the principal components in relation to the observations. We presented the factors loadings in a supplementary file (Table S1 and Table S2) and as correlation biplot on the Fig.2 and 3. We also tested if the new synthetic variables (which are equal in number to the original set) was able to ordinate significant differences in the data

  1. Response to Comments on the Quality of English Language- English does not require any improvements

Response 1: Thank you

  1. Additional clarifications

[No]

Reviewer 3 Report

Comments and Suggestions for Authors

This study investigated the impacts of two ant species on soil properties and associated differences in the soil microarthropod communities relative to soil without ant nest structures nearby. This particular study was done in a more urban setting, but it is valuable for both conservation work within an urban setting but also for general knowledge of how ant nesting structures communities. I really enjoyed reading this manuscript, with most of the central concepts laid out well for the reader. Minor comments below.

Expand a little on the conclusions in the abstract.

Line 64: Add a more detailed description of “ecological filters” for a more general readership.

Line 122: Do they resemble natural communities in plant community diversity and structure? I normally consider urban greenspaces as having less species and structural diversity than natural systems as well as having more non-native plant species.

Lines 168-167: Was this repeated sampling over this period or were the samples collected randomly from the sites one time each during this period? Given lines 211-212, I assume it was two sampling events per colony?

Tables 1 and 2: I wonder if p-valuea column should be moved to before the column for LN+LF means.

 Table S2: Moisture (%) Axis 1 has a comma instead of a period in the number.

Table 3: There is an extra decimal in the value for “Explained variation” for the first row of data.

Lines 364-365, 388 and Tables S4 and S5 mention “Richness S” and “Diversity H’”. Please, define these statistics and their calculations in the methods. Similarly, “Shannon diversity” is mentioned for first and only time in line 487.

Line 472: Are the average colony sizes and underground nest structures of these two species similar? Some differences are mentioned in lines 127-128, but I’m curious if there is overlap in other attributes that could contribute to lack of differences between the communities associated with either ant species.

Line 560: The ending sentence of the conclusions is very vague.

Author Response

Response to Reviewer 3

This study investigated the impacts of two ant species on soil properties and associated differences in the soil microarthropod communities relative to soil without ant nest structures nearby. This particular study was done in a more urban setting, but it is valuable for both conservation work within an urban setting but also for general knowledge of how ant nesting structures communities. I really enjoyed reading this manuscript, with most of the central concepts laid out well for the reader. Minor comments below.

Thank you very much for your positive opinion about our work. Please find the detailed responses below and the corresponding revisions/corrections highlighted/in track changes in the re-submitted files.

Response to comments and suggestions

Comments 1:

Expand a little on the conclusions in the abstract

Response 1:

We've gone over the abstract again, made the conclusions longer, but kept to the same number of words the editors asked for, which is 200 or under.

We corrected as follows:

Abstract Organisms that physically modify their environment, known as ecosystem engineers, can influence resource availability, species interactions and the structure of soil communities. However, the specific effect of ecosystem engineers like ants on the abundance and diversity of non-engineering soil organisms remains understudied. To address this knowledge gap, we conducted a survey of a multi-taxon belowground community of soil microarthropods—Collembola, Mesostigmata, Oribatida and Actinedida—in urban areas, comparing nest mounds of the ant species Lasius niger (Linnaeus, 1758) and Lasius flavus (Fabricius, 1782) with areas without ant-nesting activity (control). We hypothesised differences in abundance and distribution patterns of different soil microarthropod taxa between ant mounds and the control soil. We also hypothesised that ant-induced soil disturbance is species-specific, and may result in different patterns of diversity and composition of soil microarthropod assemblages within trophic levels, such as among detritivores (e.g. Collembola) and predators (e.g. Mesostigmata, which are mainly free-living predatory species in soil litter habitats). Our results reveal how ecological filters shape different soil microarthropod groups’ responses to ant-driven changes in their environment. As we expected, soil disturbance caused by ant nest-building activity significantly influenced the abundance, distribution patterns and diversity of soil microarthropods, especially in the assembly of detritivorous—but not predatory—guilds of soil microarthropods.”

Comments 2:

Line 64: Add a more detailed description of “ecological filters” for a more general readership.

Response 2

The section on understanding the concept of ecological filters has been expanded.

“These structures have specific physical, chemical and biological properties [6], and are therefore considered to be important drivers of soil diversity and ecological filters (dispersal, abiotic and biotic factors which filter species into local communities), potentially influencing the occurrence and distribution of other soil species, including microarthropods”.

Comments 3:

Line 122: Do they resemble natural communities in plant community diversity and structure? I normally consider urban green spaces as having less species and structural diversity than natural systems as well as having more non-native plant species

Response 3

There is indeed such a general assumption, but in cities, including the Warsaw metropolitan area, there are areas where remnants of natural plant communities have been preserved and management is less intense. As the fragment was unclear, we corrected it as follows:

“..Nevertheless, urban green spaces mostly that resemble natural plant communities (e.g. [43]) and are commonly colonised by ants.”

Comments 4

Lines 168-167: Was this repeated sampling over this period or were the samples collected randomly from the sites one time each during this period? Given lines 211-212, I assume it was two sampling events per colony.

Response 4

Thank you, we explain: 1 sample in colony L.flavus; 1 sample in the control; 1 in L.niger; 1 in the control. The sampling was repeated twice during spring and autumn.

We have corrected the fragment as follows: In total, 72 soil samples from ant mounds (36 from L. niger and 36 from L. flavus) and 72 36 soil samples from grasslands (control microhabitat) were collected twice during the growing season (from June and to September 2015).

Comments 5

Tables 1 and 2: I wonder if p-valuecolumn should be moved to before the column for LN+LF means.

Response 5

We believe that the table caption clearly indicates that column with p-valuesa shows the difference between L. niger and L flavus, and that column with p-valuesb shows the difference between LN+LF and the control.

Comments 6

Table S2: Moisture (%) Axis 1 has a comma instead of a period in the number

Response 6

Thank you, we corrected

Comments 7

Table 3: There is an extra decimal in the value for “Explained variation” for the first row of data.

Response 7

Thank you, we are making improvements to correct value - 68.5

Comments 8

Lines 364-365, 388 and Tables S4 and S5 mention “Richness S” and “Diversity H’”. Please, define these statistics and their calculations in the methods. Similarly, “Shannon diversity” is mentioned for first and only time in line 487.

Response 8

The section titled 'Material and Methods' has been updated to include additional information on diversity parameters which we used:

“..Differences in soil physicochemical properties and the abundance of soil microarthropod taxa (Collembola, Oribatida, Mesostigmata and Actinedida) between soil from mounds of the two soil-nesting ant species L. flavus and L. niger and from control microhabitats (without ant-nesting activity), as well as between L. niger and L. flavus mounds, were compared using a two-sided Mann–Whitney U test. Differences in basic diversity parameters, such as species richness (S) and Shannon's diversity index (H'), of Collembolan and Mesostigmata assemblages between L. niger and L. flavus ant mounds were also compared using a two-sided Mann–Whitney U.

We have also made corrections in Tables S4 and S5 and in lines 364-365, 388.

Line 364 – 365 “..The effect of L. flavus and L. niger ant mounds on the mean ranks of collembolan abundance (A) was not significant (Table 2); species richness (S) and Shannon’s diversity index (H') were also not statistically significant (two-tailed Mann–Whitney U test (N = 18, p > 0.05 in all cases).

388 as follows:

The effect of L. flavus 386 and L. niger ant mounds on the mean ranks of Mesostigmata abundance (A) was not significant (Table 2), whereas the species richness (S) and Shannon’s diversity index (H') were also not statistically significant (two-tailed Mann–Whitney U test (N = 18, p > 0.05 in all cases).

Comments 9

Line 472: Are the average colony sizes and underground nest structures of these two species similar? Some differences are mentioned in lines 127-128, but I’m curious if there is overlap in other attributes that could contribute to lack of differences between the communities associated with either ant species

Response 9

Thank you for pointing that out. We have corrected the section of our discussion you mentioned.

It is evident that the average colony size of Lasius niger and Lasius flavus does not correspond precisely. Both species form large colonies, L. flavus tends to form larger colonies as they are less aggressive and allow the formation of colonies containing more than one queen in mature nests, whereas L. niger are strictly monogynous. Notwithstanding, both species are sympatric and they nest in same habitat type and, in many occasions, very close to each other, therefore suggesting the existence of some similarities that allowing similarities in the associated communities. However, the extent to which the microhabitat of the soil changed due to the bioturbation activity of these two species clearly overlapped, making it impossible to distinguish of their impact on the soil microarthropod communities.

We corrected the fragment as follows:

Nevertheless, it is noteworthy that the nest mounds of L. niger and L. flavus did not show species-specific differences in the abundance and occurrence of higher soil microarthropod taxa, except for Mesostigmata, which occurred at much higher densities in the mounds of L. niger (Table 2) although both species differ in their ecological characteristics [45]. This scenario may be related to the different impact of the two species on soil properties due their sympatric coexistence and nesting in a disturbed habitat, such as urban green space. The differences observed in the impact of these two ant species  ecological characteristics on soil properties in urban environment: L. niger mounds were slightly less warm and moist and more fertile than L. flavus mounds (Table 1. Table S2 Figure 3b) did not cause any notable changes in soil microarthropod communities. However, T the differences in Mesostigmata abundance between L. flavus and L. niger mounds revealed in our study are in good agreement with published results showing that moisture and temperature within ant mounds are important for soil microarthropods [36].

Comment 10

Line 560: The ending sentence of the conclusions is very vague

Response 10

Thank you

  1. Response to Comments on the Quality of English Language- English does not require any improvements

Response 1: Thank you

  1. Additional clarifications

[No.]

Round 2

Reviewer 1 Report

Comments and Suggestions for Authors

I thank the authors for their answers. But still, you need to include the publications I have indicated in the introduction.

Author Response

Thank you very much for taking the time to review this manuscript. Please find the detailed responses below and the track changes in the re-submitted files

Response to comments and suggestions

Comments 1:

But still, you need to include the publications I have indicated in the introduction.

Response 1:

Please find the changes in the re-submitted files